

# Internet of Things-based sustainable environment management for large indoor facilities

Muhammad Hanif Lashari[1], Sarang Karim[2], Musaed Alhussein[3], Ayaz Ahmed Hoshu[4], Khursheed Aurangzeb[3] and Muhammad Shahid Anwar[5]

[1] Department of Electrical and Computer Engineering, Iowa State University, Ames, United States
[2] Department of Telecommunication Engineering, Quaid-e-Awan University of Engineering, Science & Technology, Nawabshah, Pakistan
[3] Department of Computer Engineering, College of Computer and Information Sciences, King Saud University, Riyadh, Saudi Arabia
[4] Department of Electronic Engineering, Quaid-e-Awam University of Engineering, Science and Technology, Larkana Campus, Pakistan
[5] Department of AI and Software, Gachon University, Seongnam-si, South Korea

Corresponding author
Sarang Karim,
sarangkarim@quest.edu.pk

## ABSTRACT

Due to global warming and climate change, the poultry industry is heavily impacted, especially the broiler industry, due to the sensitive immune system of broiler chickens. However, the continuous monitoring and controlling of the farm's environmental parameters can help to curtail the negative impacts of the environment on chickens' health, leading to increased meat production. This article presents smart solutions to such issues, which are practically implemented, and have low production and operational costs. In this article, an Internet of Things (IoT) based environmental parameters monitoring has been demonstrated for the poultry farmhouse. This system enables the collection and visualization of crucially sensed data automatically and reliably, and at a low cost to efficiently manage and operate a poultry farm. The proposed IoT-based remote monitoring system collects and visualizes environmental parameters, such as air temperature, relative humidity (RH), oxygen level ($O_2$), carbon dioxide ($CO_2$), carbon monoxide (CO), and ammonia ($NH_3$) gas concentrations. The wireless sensor nodes have been designed and deployed for efficient data collection of the essential environmental parameters that are key for monitoring and decision-making process. The hardware is implemented and deployed successfully at a site within the control shed of the poultry farmhouse. The results revealed important findings related to the environmental conditions within the poultry farm. The temperature inside the control sheds remained within the desired range throughout the monitoring period, with daily average values ranging from 32 °C to 34 °C. The RH showed slight variations monitoring period, ranging from 65% to 75%, with a daily average of 70%. The $O_2$ concentration exhibited an average value of 17% to 18.5% throughout the monitoring period. The $CO_2$ levels showed occasional increases, reaching a maximum value of 1,100 ppm. However, this value was below the maximum permissible level of 2,500 ppm, indicating that the ventilation system was effective in maintaining acceptable $CO_2$ levels within the control sheds. The $NH_3$ gas concentration remained consistently low throughout the duration, with an average value of 50 parts per million (ppm).

## INTRODUCTION

One of the most popular and widely consumed varieties of poultry products is chicken, which is frequently consumed as a better animal protein than red meat (*Chodkowska, Wódz & Wojciechowski, 2022*; *Ferrari et al., 2022*). Compared to other domestic animals, chickens are less expensive to raise for meat. Due to the exponential growth in human consumption, both the supply and demand for chicken meat are increasing daily across the majority of all countries (*Istiak & Khaliduzzaman, 2022*). This is due to the high-quality protein, low cholesterol, and low saturated fat content in poultry meat. Due to the wide range of recipes available, poultry meat is highly sought-after at restaurants, hotels, fast food restaurants, parties, *etc.* The usage of chicken in daily meals has also raised the profitability of the chicken industry. In fact, the production demand has been increasing with the passage of time. poultry meat can be obtained from two farm methods, one is open-shed poultry farms (conventional farms with the natural environment) and another one is controlled-shed poultry farms (modern farms with a controlled environment). Control-shed poultry farms are more productive and less time-consuming as compared to conventional ones. The amount of chicken meat produced is strongly correlated with the environment in which the chicks are grown up, the length of time it takes to raise the chicks, and the intensive care practices used on the farm (*Hafez & Attia, 2020*).

Lack of care at the farmhouses leads to a variety of health problems in the chickens, including breathing problems, digestive problems, and behavioural abnormalities in the control shed. In addition, broiler chicks are more sensitive than breeder chicks (*Curtin, Daley & Anderson, 2014*). Because high temperatures in the control shed might result in the usage of a lot of water, the environment must be carefully monitored (*Mahale & Sonavane, 2016*). Additionally, more trips and energy spent on cooling pads, more medication needed in case of illness, and more visits require high expenditures and biosecurity hazards. Ultimately, the current manual operation of a poultry farm's environmental components may lead to unhealthy food for consumers, higher costs, and slower business growth. There are several important environmental parameters, like temperature, relative humidity (RH), oxygen ($O_2$) level, and the concentration of carbon dioxide ($CO_2$) should be maintained in the shed. Additionally, ammonia gas ($NH_3$) concentration and carbon monoxide (CO) level must be correctly maintained for healthy poultry birds (*Raghudathesh et al., 2017*). Furthermore, appropriate management of these environmental parameters will also ensure overall environmental sustainability for the farm and the world.

The basic motivation is the role of digital technologies in the development of sustainable management of poultry farms and the growing research attention in the field of smart

farming. On the other hand, the rapid population growth has spurred the desire for more food, especially the production of chicken meat, which is in great demand. This is related to the increased requirement for using natural resources more sustainably, such as water, food, and other resources. According to the United Nations Food and Agriculture Organization (*Janczak & Riber, 2015*), digital innovation in agriculture has a huge potential to reduce poverty and hunger while simultaneously reducing the consequences of climate change (*Wijerathna-Yapa & Pathirana, 2022*). All facets of the agro-food production chain will change as a result of digitization because, the processing of enormous amounts of data in real-time enables more effective operations, larger economic returns, more environmental advantages, and better working conditions in the field (*Bolfe et al., 2020*).

## Research motivation

The use of modern technological instruments in poultry farming has already provided a significant gain in the poultry industry. Moreover, the growing connectivity in the rural environment and greater integration of data from sensor systems, remote sensing, equipment, and smartphones have paved the way for new concepts. More recently, the term "Smart Farming" has been applied to a trend that stresses the use of information and communication technologies in the digital farm management cycle, utilizing modern technologies like the Internet of Things (IoT), Wireless Sensor Networks (WSN), cloud computing, artificial intelligence, and big data (*Shaikh et al., 2022a*; *Rahu et al., 2022*; *Shaikh et al., 2022b*; *Wolfert et al., 2017*; *Karim & Shaikh, 2017*).

To help ensure that poultry is raised in a healthy environment and has a favourable influence on consumer health, we are providing a smart environment management system for poultry farms in this article. The system will help poultry managers to reduce harmful environmental elements to a bare minimum by using remote visualization and live monitoring. The proposed IoT-based smart management system will also promote sustainable business growth promote the sustainable use of critical environmental resources. The novelty of this work lies in its practical implementation of IoT technology for continuous monitoring of environmental parameters in poultry farms, its cost-effectiveness, customized sensor nodes, comprehensive data collection, and the valuable insights gained from the deployed system. By addressing the challenges specific to broiler chicken health and meat production in the context of global environmental changes, the research contributes significantly to the field of sustainable agriculture and IoT-based solutions for animal welfare and productivity. The main contributions of this article are outlined below, and are based on our previous work (*Lashari et al., 2018*):

- Practical implementation of IoT-based sustainable environment management for large indoor monitoring system in a poultry farmhouse.
- Low-cost and efficient management of poultry farms through real-time data collection and visualization.

- Provides findings on environmental conditions within the poultry farm, including air temperature, relative humidity (RH), oxygen level ($O_2$), carbon dioxide ($CO_2$), carbon monoxide (CO), and ammonia ($NH_3$) gas concentrations.
- Discusses the limitations of the proposed work and suggests future recommendations to enhance this work.

The remaining sections of the article are organized as follows. "Background" discusses the background followed by existing work on IoT-based poultry farm management and monitoring system. "Proposed System" provides details about the proposed system along with software and hardware implementations. "Results and Discussion" discuss the findings for the key parameters as mentioned earlier. "Limitations and Future Roadmap" highlights the limitations of this study and also provides a future roadmap. The article is concluded in "Conclusions".

## BACKGROUND

This section provides a detailed discussion of previous studies on poultry farm environment management. In addition, we also discuss the importance of the poultry farm environment.

### Poultry farm environment

Different researchers have highlighted the issues in the poultry farmhouse environment and highlighted the importance of environmental parameters.

*Arhipova et al. (2021)* explored the utilization of technological innovations in poultry farming to enhance productivity and mitigate greenhouse gas emissions. The authors analyze existing smart poultry management systems that enable optimal feeding processes intending to reduce ($CO_2$) and ($NH_3$) emissions. The study examines various technologies and research directions related to smart poultry management systems, focusing on their potential to improve productivity, predict health and welfare outcomes, and reduce greenhouse gas emissions. It provides valuable insights into the implementation of smart technologies in the poultry industry to optimize feeding processes, enhance productivity, and minimize greenhouse gas emissions. Additionally, the article emphasizes the research gap in multi-criteria decision-making and discusses the challenges associated with the implementation of such a smart platform.

*Janczak & Riber (2015)* explained how the environmental circumstances in which poultry are bred directly affect the productivity and well-being of hens and broiler chickens. *Gebregeziabhear & Ameha (2015)* noted that the thermal comfort range of chickens changes mostly based on their age, it is preferred that the temperature in the shed stay within the prescribed range for each age group during the breeding time. Stress from heat or cold can affect a flock's mortality rate, welfare, and food intake and can make the birds more susceptible to illness. *Najafi et al. (2015)* described as being between 50% and 70% relative humidity within the shed. Low relative humidity accelerates the rate of heat loss through evaporation, causing dryness of their mucous membranes and airways. On

the other hand, high temperatures and high relative humidity may deteriorate the performance of broiler chickens.

*Naseem & King (2018)* have shown that the biggest source of environmental contamination from chicken farming is ammonia ($NH_3$). When birds eat protein, they create uric acid, which under favourable circumstances is ultimately transformed into $NH_3$, pH, temperature, moisture content, kind of litter, bird age, manure age, RH, and Ventilation Rate (VR) are all factors that boost output. In chicken buildings, $NH_3$ emissions and concentration are influenced by VR, as well as by the seasons. The ecosystem, environment, and health of both people and birds are negatively impacted by $NH_3$. $NH_3$ exposure level should be less than 10 parts per million (ppm), although levels of up to 25 ppm are still safe. *Sousa et al. (2016)* in the traditional broiler chicken breeding technique, the birds are kept in a masonry shed with a floor that is often made of wood and covered in broiler litter. This is an example of the physical environment. Moisture from bird excrement may be absorbed by broiler litter and other items in the vicinity.

*Oro & do Prado Guirro (2014)* have addressed the traditional setup for the chicks. The majority of the hens used for egg production in the intensive system are caged, and excreta is frequently stored for many weeks in the shed. *Nkoa (2014)* has explained how $NH_3$ is produced. As a result, ($NH_3$) can be produced in both methods of chicken husbandry when microbes break down organic waste found in excreta and/or litter. *David et al. (2015)* highlighted the dangers of $NH_3$. $NH_3$ is a hazardous gas, and hence it is advised that its airborne concentration be monitored and kept below 10 ppm to prevent harming bird and crew health as well as animal performance.

*Jácome, Rossi & Borille (2014)* described how improved egg quality is affected by light. Due to their photosensitivity, birds can be affected by exposure to daily light periods that are shorter or longer than what is ideal, which can affect their ability to reproduce. Therefore, in light of considerations, it is necessary to continuously monitor the environmental factors in the control sheds. This procedure is crucial because it enables poultry producers to act quickly when it becomes clear that the chicks' habitat needs to be improved. The standard parameters for poultry farmhouses within the control shed from weeks 1 to 4 are given in Table 1 (*Ammad-Uddin et al., 2014*).

## Previous studies

Some researchers have presented their work with different techniques for IoT implementation in poultry farms and we will discuss them in this section.

*Archana, Uma & Babu (2018)* suggested that Raspberry Pi3 could be used to monitor and control poultry farms by measuring environmental parameters such as temperature, humidity, and air quality using four sensors (smoke, temperature, humidity, and gas). The watering system is also controlled by a float sensor, and all of the sensors are integrated with the Raspberry Pi3, which can collect and analyze data.

*Zhang & Chen (2020)* created an Android-based intelligent portable terminal to enable real-time and remote wireless monitoring for indoor hazardous gas concentration, illuminance, and environment temperature and humidity farm.

**Table 1 Standard ranges of physical parameters within poultry farm** (*Ammad-Uddin et al., 2014*; *Czarick & Fairchild, 2007*).

| Physical parameters | First week | Second week | Third week | Fourth week |
|---|---|---|---|---|
| Temperature | 34–32 °C | 32–30 °C | 30–28 °C | 29–27 °C |
| Relative humidity | 80–60% | 80–60% | 80–60% | 80–60% |
| Carbon monoxide | Must be 0 ppm | <10 ppm | <25 ppm | <50 ppm |
| Carbon dioxide | <1,000 ppm | <2,000 ppm | <2,500 ppm | <2,500 ppm |
| Ammonia level | Near to 0 ppm | <10 ppm | <20 ppm | <25 ppm |
| Oxygen level | 16–18% | 16–18% | 16–18% | 16–19% |

*Thomas et al. (2020)* presented a sensor technology and microcontroller-based system that monitors internal parameters within the farm. The temperature and humidity (measured using a DHT11 sensor) are monitored using a microcontroller, and a fan or heater is activated as needed to maintain optimal conditions.

*Du et al. (2021)* offered a gateway design approach based on low-power Bluetooth technology that may be used to monitor a breeding environment. The implementation of the priority queue and feedback mechanism into the gateway design successfully addresses the issues of Bluetooth devices' limited communication range and low efficiency.

*Kodali, Yerroju & Sahu (2018)* discussed various wireless communication technologies used in the IoT. One example given was the use of long-range radio (LoRa) to transmit sensor data, such as temperature, humidity, and soil moisture, from a transmitter node to a receiver node. The receiver node, which is equipped with Wireless Fidelity (Wi-Fi), uses Message Queuing Telemetry Transport (MQTT) services to check the data in the IBM Watson IoT platform and store it in the IBM cloud database service.

*Choukidar & Dawande (2017)* described involves the use of a combination of wireless sensors and a General Packet Radio Services (GPRS) network to control and monitor environmental parameters in a poultry farm. The identified and defined environmental parameters are necessary for the optimal growth of chickens. An integrated solution using a WSN and GPRS network is proposed to transmit data and maintain a detailed record of the status of environmental conditions in poultry farms on a webpage.

*So-In, Poolsanguan & Rujirakul (2014)* focused on blending the use of mobile devices and sensor network technology to remotely regulate and monitor environmental factors in a chicken poultry farm. By sending Short Messaging Services (SMS) to the owner's registered cellphone number, this system enables the owner to keep track of numerous environmental indicators including changes in temperature and humidity. When the system does not receive a command at a specific time, it will automatically start the needed action. The owner can start a required action by sending a message back to the system to execute. As a result, the system's architecture offers an effective, automated, and novel approach to monitoring chicken farms.

*Bang et al. (2014)*, described the use of a Light Emitting Diode (LED) smart lighting control system in poultry farms to address the low energy efficiency and high power consumption of traditional systems using incandescent bulbs for illumination. The

proposed smart control system has several benefits, including improved energy efficiency, increased illumination control ranges (up to 10 times more than the existing system in the poultry farms), and the ability for a farmer to remotely manage their poultry farms through real-time environmental monitoring using a Personal Computer (PC) or smartphone.

*Dong & Zhang (2009)* advised the need for real-time monitoring of poultry farms in the environment. Thus, an online monitoring system based on the ZigBee module was developed. It offers a network of real-time monitoring systems. The ZigBee-based CC2430 from TI serves as the node controller, data receiver, data transmitter, and control node for the network of real-time monitoring systems. Different environmental characteristics are detected by ($CO_2$) sensors using TGS4161, and temperature and humidity sensors using SHT75. By analyzing the system's data transmission, the ZigBee protocol stack was made simpler, and the system's data transmission protocols and communication formats were created.

*Olaniyi et al. (2014)* highlighted that low production costs and high levels of employee participation on chicken farms might result in poor profit and a low return on investment. This research was motivated by problems in the inadequate chicken feeding system, and the study led to the creation of a clever fuzzy logic-based system that could take the place of the poultry workers by providing water and food for the birds at predetermined intervals. The device is meant to sense the level of the water and feed and intelligently dispense it in response to fluctuations in the level as the hens consume the food and water. The burden on the poultry attendants is lessened by this approach, which also enhances cost savings and produces strong returns on investment for the poultry farming system (*Olaniyi et al., 2014*).

*Muttha et al. (2014)* noted that poultry farming is currently done manually, limiting the benefits that farmers can achieve. In the past, it was common to feed chickens whole cereals as scratch feed or as part of a complete diet. However, with the growth of large-scale poultry production, automatic feeding systems using full-fed complete diets have become the dominant choice. One of the key tasks in poultry farming is controlling and monitoring environmental parameters to ensure the proper care of chickens, which can be achieved through the use of a sensor-based system. Automation using a programmable logic controller (PLC) can be used to create an environmentally controlled poultry shed that is monitored 24 h a day to improve efficiency, reduce the need for human labour, and minimize human error.

*Boopathy et al. (2014)* outlined the measurement of various environmental factors at the poultry farm, such as temperature change, humidity fluctuation, water level management, and valve adjustment. The calibration of the temperature sensor was carried out using a two-point formula. The level sensor can gauge and output the fuel level in the generator.

*Okada et al. (2013)* developed a low-power wireless sensor node for regular monitoring of activity for animal health care. The measurement of body temperature is sufficient for health care purposes and effective at reducing power usage. The proposed approach involves an ultra-low-power technique for regular activity measures using a customized large-scale integration (LSI) circuit with an estimated power usage of 320 nW in the

standby state and micro-electromechanical systems (MEMS) piezoelectric micro-cantilever.

*Huang et al. (2023)* propose an intelligent algorithm called Energy Control and Computation Offloading (ECCO) to address the challenges of real-time data processing and minimizing the Age of Information (AoI) in the Industrial Internet of Things (IIoT) systems. The authors focus on optimizing energy consumption while satisfying AoI constraints by leveraging deep reinforcement learning (DRL) techniques. They conduct extensive analyses on real-world IoT datasets, develop queueing models for IoT devices and edge servers, and formulate a dynamic optimization problem using Markov Decision Process (MDP). The ECCO algorithm is designed to adapt to the dynamic IIoT environment and is evaluated through experimental simulations. The results demonstrate its superiority over existing DRL and non-DRL algorithms in terms of minimizing AoI and energy consumption. Overall, the article presents a solution that combines energy control, computation offloading, and AoI optimization to enhance the performance of IIoT systems in industrial settings.

## Why LoRaWAN?

Due to the massive high-speed data coverage they provide, cellular networks are extremely popular. On the other hand, IoT applications do not have an immediate requirement for high-speed data. However, the battery life of devices accessing cellular networks is frequently relatively short. There are some flaws with the network coverage as well. Mesh networks identical to ZigBee, however, are frequently employed in home automation. This is a result of the low to medium-distance performance it provides. However, mesh networks catastrophically fail over extended distances (a few kilometres). Although Bluetooth has a reasonably good data rate, its range severely limits its use. Because of its limited coverage, Bluetooth is a poor choice for long-range IoT applications.

Among all wireless technologies, the IEEE802.11 WLAN (Wireless Local Area Network) standard is undoubtedly one of the most popular. This is the result of its high bandwidth and data rate alone. On the other hand, its range has some limitations and it puts strain on the battery. Wi-Fi-enabled devices frequently need to be close to the Wi-Fi access point in order to link successfully and they have a short lifespan. However, because the sheds in poultry farms are far apart from one another, adding more access points or routers due to Wi-Fi use can raise the cost of network growth. Additionally, because of the high working frequencies (2.4 and 5 GHz), it is difficult for the waves to pass through obstructions (*Kodali, Yerroju & Sahu, 2018*).

Other options such as NarrowBand-Internet of Things (NB-IoT) and Sigfox are suitable for these types of applications (*Shaikh et al., 2022a*). But these technologies are not fully functional all around the globe. Therefore, NB-IoT is a licensed protocol and Sigfox has low bandwidth. Neither has coverage in developing countries, such as Pakistan. These issues can be resolved by utilizing the IoT as a network of web-connected devices that communicate with one another (*Want, Schilit & Jenson, 2015*). Finally, for these types of applications, the solution perfectly fits in the Low Power Wide Area Networks (LPWAN). The primary shortcomings of existing wireless communication technologies, such as

limited bandwidth, higher power consumption, and coverage limitations, are solved by IoT applications using LPWAN. Designing the IoT applications, the LoRaWAN (Long Range Wide Area Network) protocol or LoRa in the LPWAN domain offers greater benefits including scalability, security, and resilience (*Kodali, Yerroju & Sahu, 2018*).

LoRaWAN technology is rapidly developing, and IoT-driven applications will use it in many different contexts. It is well suited for those applications where the data rate is not a priority. LoRa (*Griva et al., 2023*) is an unlicensed protocol as compared to the Cellular network and NB-IoT. LoRa with Cellular or NB-IoT can solve this problem. Farms are in remote areas where internet connectivity is limited, only 2G coverage is available in remote areas. The LoRa wide area network protocol sometimes referred to as LoRa, offers benefits including compatibility between LPWAN networks, simple installation, increased affordability, flexibility, scalability, bi-directionality, security, and encryption. A significant fraction of the billions of devices needed to foresee the Internet of Things is supported by LPWAN technology. For anticipated volumes of sensor-based Internet of Things applications where just a limited quantity of sensed data needs to be carried over vast distances, this LPWAN technology delivers an effective battery lifespan. The cost for such IoT applications LoRaWAN is built in the LPWAN domain to improve range and battery longevity (*Lavric & Popa, 2017*).

# PROPOSED SYSTEM

## Software implementation

The software implementation for IoT-based sustainable environment management for large indoor poultry farms requires careful consideration to measure the environmental parameters. The overall steps of the software implementation for IoT-based sustainable environment management for large indoor poultry farms are depicted in Fig. 1.

**Start:** This is the starting point of the software implementation. It indicates the beginning of the process.

**Initialize the IoT devices:** The code begins by initializing the IoT devices and sensors used in the poultry farm system. This step ensures that all the necessary hardware components are set up and ready for data collection. It includes establishing connections with sensors, configuring communication protocols, and ensuring the proper functioning of the devices.

**Data collection and transmission:** This step involves configuring the data collection and transmission protocols. It ensures that the sensors are properly connected to the IoT network and are capable of transmitting data to the monitoring system.

**Set threshold values:** In this step, we define the desired threshold values for each parameter, such as temperature, RH, ammonia, carbon monoxide, carbon dioxide, and oxygen. These thresholds serve as reference values for determining whether the parameter readings are within the acceptable range or not.

**Readings and visualization:** The system reads the sensor values using appropriate methods or functions. This could involve querying the sensors for specific data or receiving data through a communication interface. The retrieved sensor values are then displayed on an LCD screen for real-time visualization and monitoring.

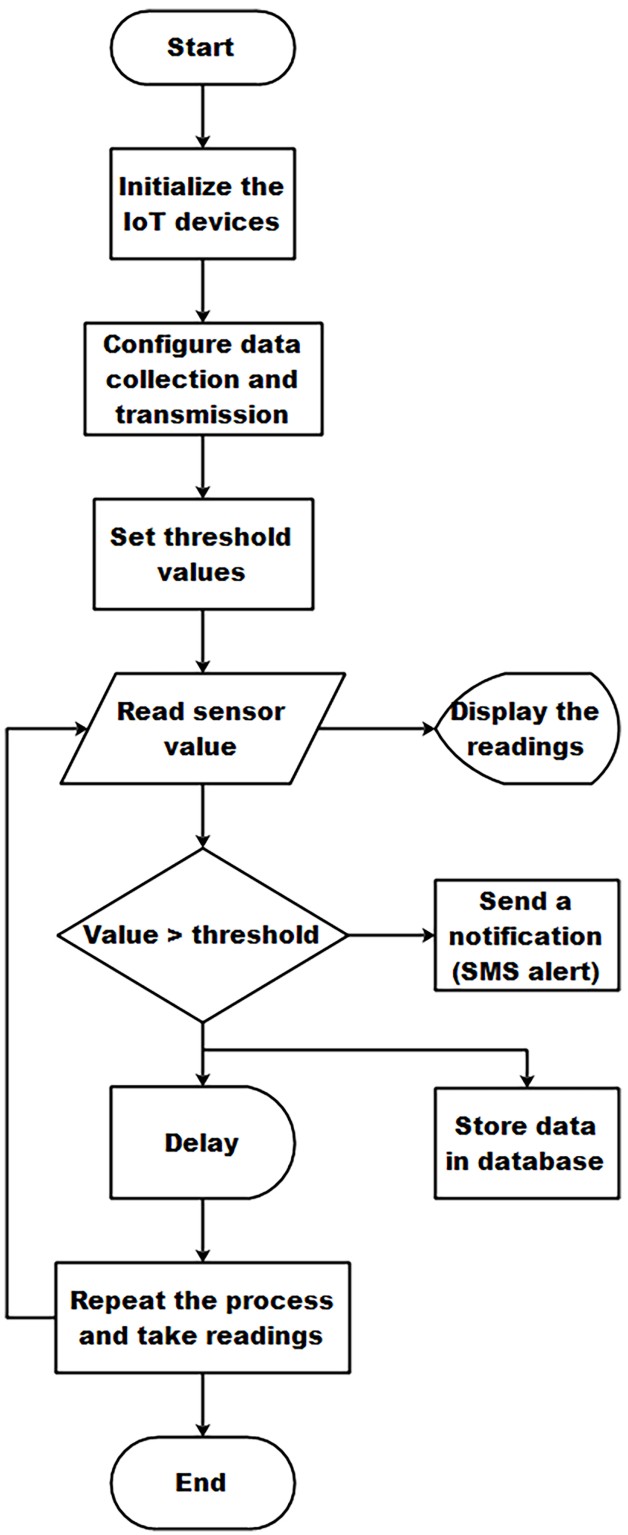

**Figure 1  Flowchart of the proposed IoT-based sustainable environment management.**

**Generate a notification (SMS alert):** This step involves checking if the sensor value exceeds the predefined threshold. If the value exceeds the threshold, it indicates an abnormal reading. In such cases, the system generates a notification, such as an SMS alert, to notify the relevant stakeholders or responsible personnel about the abnormal condition for immediate action.

**Store the data in the database:** The system stores the collected sensor data in a database for future reference and analysis. This includes recording the sensor values along with their associated timestamps to create a historical dataset for trend analysis and decision-making.

**Delay:** This step introduces a delay or pause in the execution flow. It allows for a specific interval between sensor readings, which can be adjusted based on the system requirements and the desired frequency of data collection. The delay ensures efficient resource utilization and avoids excessive data collection.

**Repeat to read the sensor value:** After the delay, the flowchart loops back to the step of reading the sensor value. This enables continuous monitoring and data collection in a cyclic manner.

**End:** This is the endpoint of the software implementation. It indicates the completion of the monitoring process.

## Hardware implementation

In this article, the poultry environment monitoring system using LoRa communication is presented. The system has two types of nodes, aggression nodes, and sensor nodes. The aggression node is compromised of the LoRa gateways (*Loyse, 2017*) and works as a fog computing device that sends data to the web server. It consists of a Raspberry Pi, Global System for Mobile Communication (GSM) module, and LoRa transceiver. Sensor nodes are designed to acquire real-time data, they consist of Arduino with LoRa Transceiver and sensors.

The overall concept of the proposed system is depicted in Fig. 2. Here, we have shown three poultry farmhouses and each of them consists of three control sheds. Our system collects data from each shed and sends it to the LoRa gateways. Gateways subsequently send data using GSM technology to the cloud (AWS (Amazon Web Services), in our case) for storing and analysis purposes. This stored data not only be viewed but also monitored easily. With the help of this scheme, different poultry farmhouses can be connected and monitored. We can access multiple sites by using a single dashboard. The system is very flexible, so poultry farmhouses can be connected easily just to plug and play from any site. Different sensor nodes can be connected with a single aggression node or gateway. Since there is a single gateway in each farmhouse, a single Subscriber Identity Module (SIM) for 2G connectivity is needed. This saves network costs and fewer installation efforts are needed. This was an issue in our previous work that resulted in high expenses due to multiple GSM and SIMs as mentioned in (*Lashari et al., 2018*).

The experimental setup involved the strategic placement of sensors to ensure comprehensive coverage of the entire control shed area. The sensor placement was carefully designed to capture temperature variations in different areas of the control shed. Specifically, one sensor was positioned on the low-temperature side or cooling pad side,

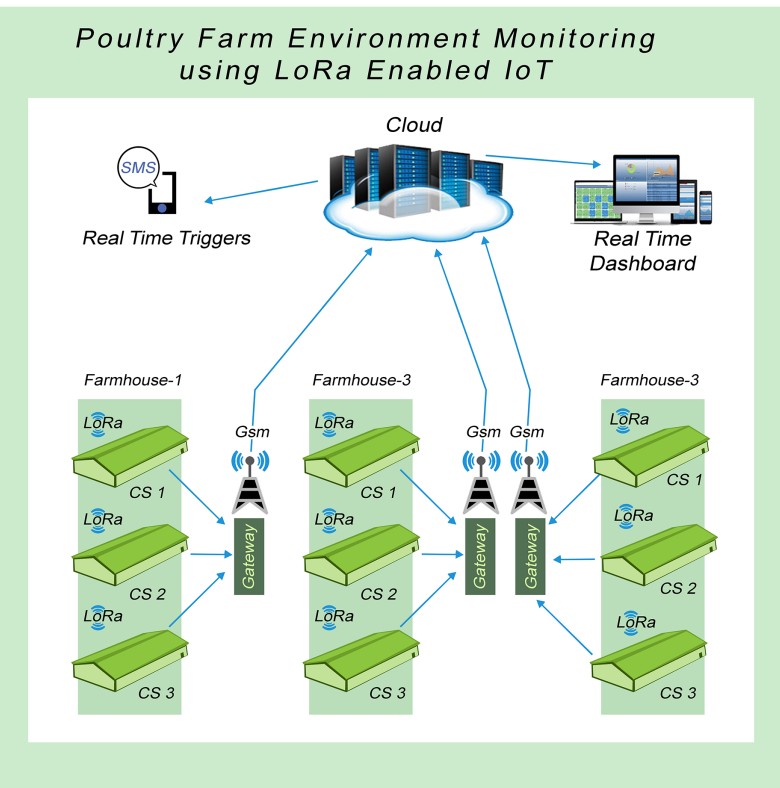

**Figure 2** System model of the proposed IoT-based sustainable environment management.

another sensor on the air exhaust side or fan side, and a third sensor at the center of the control shed. Moreover, devices should be placed close to the ground, two to three feet above the ground. So chick can not easily reach there, and more accurate data can be collected. Actual device placement, as shown in Fig. 3. Furthermore, each location within the control shed experiences distinct temperature patterns, and it is crucial to monitor and maintain a consistent environment throughout the shed.

In addition to the physical sensor placement, the study also incorporated an IoT-based infrastructure to collect and transmit the acquired data. The sensor nodes/devices utilized the LoRa communication protocol to send the collected data to a central gateway. This gateway served as a data aggregation point for further processing and analysis.

The strategic arrangement of sensors within a controlled environment is known as the "control shed." The sensors being discussed are specifically intended to measure parameters like temperature, humidity, and gas concentrations (CO, $CO_2$, and $NH_3$). This spatial arrangement is depicted in a diagram referred to as Fig. 3.

The primary purpose of this visual representation is to illustrate how these sensors are positioned within the control shed. The careful positioning of these sensor nodes/devices has been done deliberately at three distinct locations within the control shed. This strategic placement has been designed to ensure comprehensive coverage of the entire spatial area,

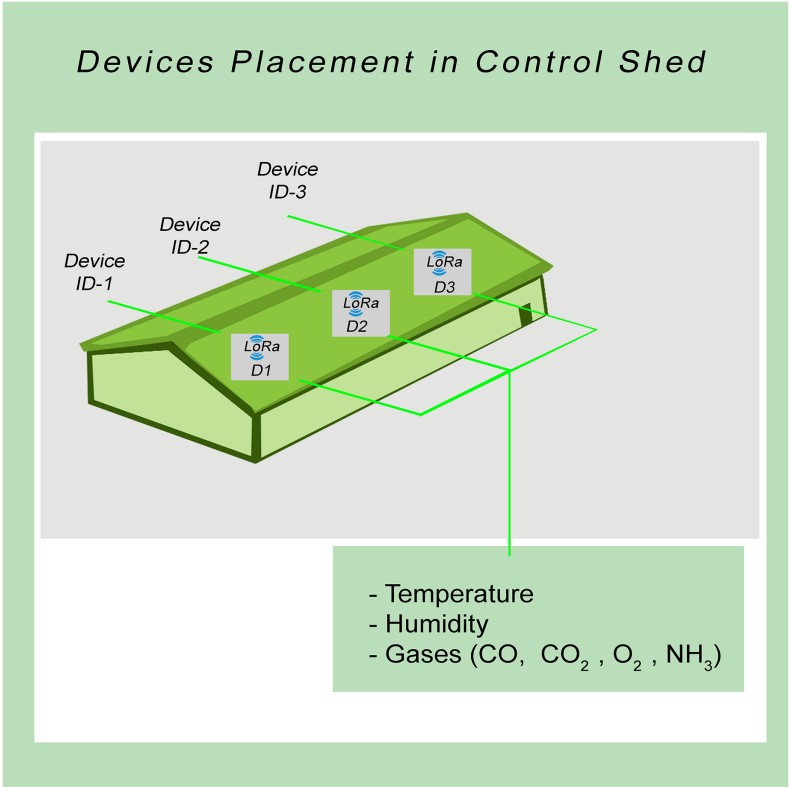

**Figure 3 Sensor nodes placement.**

allowing for effective and accurate monitoring of temperature fluctuations across the entirety of the control shed.

In essence, Fig. 3 offers a detailed illustration of how sensors measuring temperature, humidity, and gas concentrations (CO, $CO_2$, and $NH_3$) are strategically situated at different points within the control shed. This arrangement has been methodically devised to provide optimal coverage, enabling the monitoring of temperature variations across the entire space of the control shed. This visual representation aids in comprehending the practical implementation and positioning of these sensors for effective data collection and analysis.

### Sensor nodes or devices

Control sheds are normally tunnel-shaped and have an average size of about $50 \times 400$ feet. Keeping these dimensions in mind, three devices are placed in each control shed. If any poultry farmhouse has three control sheds in it, at least nine devices are needed for proper data acquisition. These devices are Arduino-based with LoRaWAN communication. The devices were placed in the control shed of the poultry farmhouse for real-time data extraction and collection. The detailed block diagram of a sensor node is shown in Fig. 4.

Initially, we tested different types of sensors. In the end, we got some very good quality sensors, and we used these in our final sensor node product. The sensor node or device consists of Arduino Mega (2560), LCD (Liquid Crystal Display) $16 \times 2$ module for data

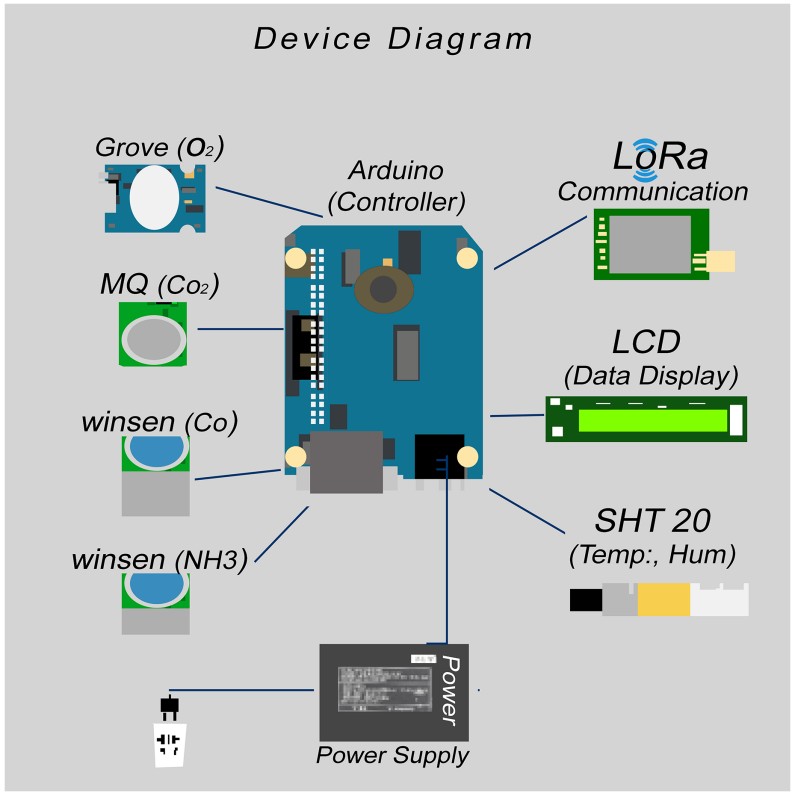

**Figure 4** Connection diagram of IoT devices.

**Table 2 List of sensors: model number, calibration and their ranges.**

| Parameter | Sensor model | Calibration method | Measurement range |
|---|---|---|---|
| Temperature and humidity | SHT20 | Factory calibrated | −40 °C to 125 °C |
| Oxygen (O$_2$) Level | Grove O$_2$ EN0322 | Periodic calibration | 0% to 30% |
| Carbon dioxide (CO$_2$) | MQ-135 | Manual calibration | 0 to 10,000 ppm |
| Carbon monoxide (CO) | Winson-ZE07-CO | Periodic calibration | 0 to 500 ppm |
| Ammonia (NH$_3$) | Winson-ME3NH3 | Periodic calibration | 0 to 100 ppm |

display, LoRa Transceiver module for data transmission, and different air quality sensors. The list of sensor name and their respective model numbers is given in Table 2.

**SHT20:** The SHT20 sensor functions as a slave in Inter-Integrated Circuit (I2C) communication. Upon request by the microcontroller, the sensor starts a process of measuring temperature and humidity. After completing the measurement, the sensor sends the data to the microcontroller and enters an idle mode to minimize energy consumption. The time required to measure temperature and humidity ranges from 5 to 30 s (*Du et al., 2017*).

**Grove O$_2$ sensor:** A Grove oxygen sensor is used to measure the concentration of oxygen as a percentage in the air. This sensor is modelled after the concept of the electrochemical cell in the original work. This sensor is adequate for detecting oxygen

levels in the environment. This works on an organic reaction module, it can give a small amount of current when exposed to air (*Boonsong et al., 2017*).

**MQ-135:** The MQ-135 gas sensor was the only available option we have for measuring the $CO_2$. Within the sealed environment, it detects $CO_2$. This gas sensor requires a voltage range of 2.5 to 5.0 V. The MQ-135 sensor is sensitive to a wide range of gases and is particularly sensitive to NH3, NOx, alcohol, Benzene, smoke, and CO2. The output is an analogue voltage that can be read by a microcontroller to determine the concentration of the target gas (*Shukla & Suman, 2021*).

**CO sensor:** An electrochemical carbon monoxide (CO) detector module, such as the ZE07-CIO, is a device used to detect the presence of CO gas in the air. It typically works by using an electrochemical cell to measure the concentration of CO gas in the air. The ZE07-CIO, specifically, is a CO sensor module that uses an electrochemical cell sensor to detect the presence of CO gas. This type of sensor is known for its high accuracy and stability, making it suitable for use in a wide range of applications, including home safety, industrial process control, and automotive exhaust systems (*Barot, Kapadia & Pandya, 2020*).

**NH$_3$ sensor:** The $NH_3$ sensor is another electrochemical sensor that determines gas concentration by measuring current. It works on the electrochemical principle and uses the electrochemical oxidation of the target gas on the working electrode inside the electrolytic cell. As per Faraday's law, the current produced in the electrochemical reaction of the target gas is directly inversely proportional to its concentration (*Sun, Palaoag & Quan, 2022*).

### Constraints and concerns

The implementation of an IoT-based sustainable environment management system in poultry farms involves the deployment of various sensors to monitor and regulate environmental conditions. These sensors play a critical role in ensuring optimal conditions for poultry health, growth, and overall farm productivity. However, there are specific concerns and constraints associated with using sensors in such environments, particularly due to potential animal interactions and their impact on measurement precision.

**(a) Animal interaction with sensors:** Poultry's natural curiosity and behavior might lead them to interact with or peck at sensors. This interaction could potentially disrupt sensor functionality or damage the sensors themselves. Steps need to be taken to design sensor enclosures or mounting strategies that minimize such interference.

**(b) Measurement precision and accuracy:** Poultry interactions or environmental factors might affect the accuracy and precision of sensor measurements. For instance, animals crowding around a temperature sensor might block proper airflow, leading to inaccurate readings. Careful sensor placement and calibration are essential to mitigate these issues.

**(c) Data noise and interpretation:** Poultry movement or interactions could introduce noise into the sensor data, making it challenging to distinguish between normal behavior and potential anomalies. Advanced data filtering and analysis techniques may be required to extract meaningful insights.

**(d) Sensor maintenance and replacement:** In an agricultural setting, sensors can be exposed to dust, debris, and moisture, potentially affecting their longevity and reliability. Regular maintenance and timely replacement of damaged sensors are crucial to ensure continuous monitoring.

**(e) Energy efficiency:** Ensuring sensors operate efficiently without causing stress to the poultry is important. Bright LED lights on sensors, for instance, might disturb the birds, affecting their behavior and health.

**(f) Integration with farm practices:** The deployment of sensors should align with the overall management practices of the poultry farm. Sensors should complement existing protocols and practices rather than disrupt them.

Addressing these constraints requires a combination of thoughtful sensor design, strategic placement, data analytics techniques, and a deep understanding of poultry behavior. Despite these challenges, IoT-based sustainable environment management holds significant promise for improving poultry farm operations and animal welfare.

### Gateway design

A single gateway is designed to collect all the data coming from different sensor nodes/devices and send this data to a cloud server using GSM. The communication between devices and the gateway is done through LoRa-based protocol. The gateway consists of Raspberry Pi, a GSM module, and a LoRa transceiver. Sensor nodes send data every 60 s to the gateway. Furthermore, Raspberry Pi acts as a fog computing device that collects data from different nodes and processes the data, stores the data within the node, and then sends it to the main server. It also sends critical information in emergence mode. Usually, Raspberry Pi receives a huge amount of sensor data, but it only sends important data, such as average values, to the cloud to minimize cloud charges/costs. It sends data after a 4 to 5 min delay (we can increase or decrease the data sending period according to requirement). Raspberry Pi also send SMS alert or notifications to the supervisor, doctor, and owner of the poultry farmhouse, allowing them to take immediate action.

In cases where the internet is temporarily unavailable at the site, the data will be stored in Raspberry Pi. Raspberry Pi can store the data of a complete flock/cycle and send data whenever it gets internet connectivity. In case of any failure in the sensor nodes, we can access data remotely through the TeamViewer software installed on Pi. This can be accessed using an id and password. The Block diagram of the pi-based gateway is shown in Fig. 5.

### Real-time deployment

We have successfully deployed a real-time IoT-based sustainable environment management for large indoor poultry farms, which is depicted in Fig. 6. This system, as shown in Fig. 6, enables continuous monitoring and analysis of various environmental parameters in poultry farms. It collects data from sensors installed throughout the farm, including temperature, humidity, air quality, and light intensity. The collected data is then transmitted wirelessly to a central server for processing and analysis. This system provides farmers with valuable insights into the environmental conditions of their poultry farms,

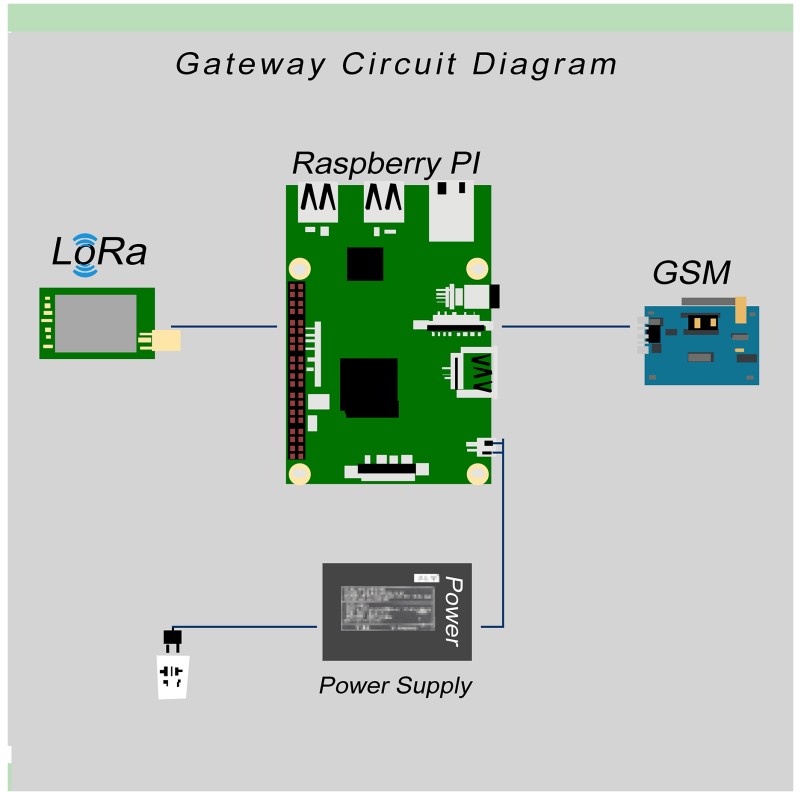

**Figure 5  Gateway circuit diagram.**

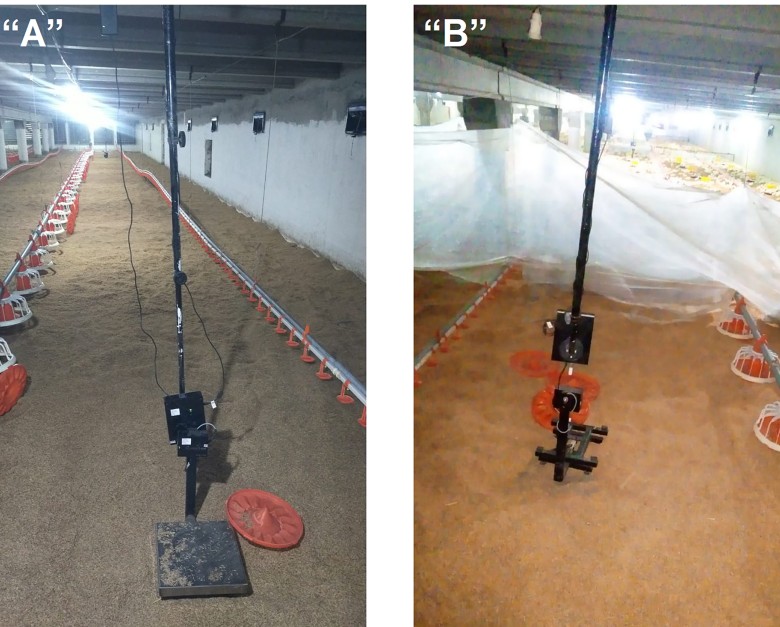

**Figure 6  Real-time deployment of IoT nodes.** (A) Scenario 1: setup (B) Scenario 2: monitoring.

allowing them to make informed decisions and take necessary actions to optimize the well-being and productivity of their poultry.

### Software application

The Poultry Environment Monitoring System (PEMS) comprises a cohesive software and hardware infrastructure. Operating under the Software as a Service (SaaS) paradigm, PEMS employs a suite of diverse software tools to ensure its functionality. These tools include:

(a) **Eclipse** (*Alizadehsani et al., 2022*): Eclipse, a widely-used integrated development environment (IDE), plays a pivotal role in facilitating the creation, modification, and management of PEMS's software components. It provides a robust platform for coding, debugging, and testing, ensuring efficient software development.

(b) **MySQL (Structured Query Language)** (*Liiv, 2021*): MySQL, a robust relational database management system, underpins PEMS's data storage and retrieval mechanisms. Employing SQL, MySQL efficiently handles the structured data that PEMS collects and processes, ensuring the system can manage poultry farm data effectively.

(c) **MySQL Workbench** (*Krogh, 2020*): MySQL Workbench is a graphical tool that complements MySQL's functionality. It aids in the design, modelling, and administration of the database schema, enabling streamlined interaction with the underlying data structure.

(d) **WildFly** (*Stancapiano, 2017*): WildFly, a potent, modular, and lightweight application server, empowers the creation of exceptional applications. Previously recognized as JBoss AS or simply JBoss, WildFly originates from JBoss and is presently nurtured by Red Hat. Crafted using Java, WildFly adheres to the Java Platform, Enterprise Edition specification, ensuring seamless implementation of enterprise-level solutions. The WildFly application server provides a runtime environment for PEMS's web-based application. It manages the execution of Java-based components, ensuring seamless communication between the application and the underlying hardware.

(e) **Java Development Kit (JDK):** The Java Development Kit is an essential toolkit for PEMS's software development. It includes the Java compiler, runtime environment, and various libraries, enabling the creation of the application's core functionality using the Java programming language.

PEMS itself serves as a dynamic web-based application, leveraging the capabilities of Amazon Web Services (AWS) for deployment. Its primary purpose is to monitor and manage diverse poultry farmhouses. Through a secure user authentication system, authorized users access the application *via* a URL. This access empowers them to monitor environmental conditions, track data trends, and make informed decisions to optimize poultry health and production within the farmhouses.

In essence, PEMS harnesses the capabilities of these software tools to deliver a comprehensive solution that enhances the monitoring and management of poultry farm environments. The seamless integration of Eclipse, MySQL, MySQL Workbench, WildFly, and JDK forms the technological backbone that empowers PEMS to provide real-time insights and actionable data for effective poultry farm management.

## RESULTS AND DISCUSSION

We conducted comprehensive monitoring of six crucial air quality parameters in the control shed of the poultry farmhouse using a range of sensors. These parameters included temperature (°C), relative humidity (RH), oxygen ($O_2$) level, carbon dioxide ($CO_2$) level, carbon monoxide (CO) level, ammonia ($NH_3$) gas concentrations, inside the shed. The deployment of these sensors took place in the control shed located in Sehwan, Sindh, Pakistan. Sensor nodes were strategically placed on-site and monitored remotely. The figures presented in the study represent the temporal variation of each physical parameter, with the magnitude on the vertical axis and time on the horizontal axis.

While the data presented reflects real-time information collected from the poultry farmhouse, it is important to note that the effectiveness and robustness of the monitoring system's algorithm were validated through extensive testing. The algorithm demonstrated its ability to provide accurate and reliable alerts and graphical representations of the data to the supervisor in the farmhouse. The supervisor found the system instrumental in making informed decisions and taking immediate actions when necessary.

Furthermore, the system facilitated the storage and analysis of data from different poultry flocks, enabling the farm owner to make informed decisions about flock management. This information was invaluable in determining the optimal conditions for starting a new flock and ensuring favourable conditions for their growth and well-being.

The integration of sensor data, algorithm validation, and practical applications highlights the effectiveness of the monitoring system in providing real-time insights and facilitating decision-making processes in the poultry farmhouse environment.

### Result

In the following section, we provide a detailed analysis of the results for each sensing parameter. The data presented represents the daily average of sensor readings collected at 30-min intervals over the course of 1 week.

### Temperature sensor

Maintaining a consistent and suitable temperature inside the control shed of a poultry farmhouse is crucial for the well-being and growth of the chicks. In order to monitor the temperature, the SHT20 sensor was employed in this system to collect temperature data, measured in degrees Celsius (°C). The first-week data is presented in Fig. 7, with the X-axis representing the dates and the Y-axis indicating the daily average temperature. Analyzing Fig. 7, it can be observed that the temperature inside the control shed remained within an acceptable range over the specified dates, with a maximum value of 34 °C, which adheres to the upper limit recommended by the standards (*Ammad-Uddin et al., 2014*). However, it is worth noting that the temperature fell below the lower limit of 30 °C for the last 2 days of the measured period. To ensure that the temperature remains within the desired range, the proposed system is designed to send an alert to the supervisor whenever the temperature drops below 30 °C on specific dates. This notification prompts the supervisor to take appropriate actions, such as activating the vents or cooling pads, to regulate the temperature and maintain the desired conditions for the chicks' comfort and well-being.

**Daily Average Temperature Level (°C)**

**Figure 7  Temperature values in Celsius.**

Looking ahead, the integration of this system with different IoT actuators will allow for fully controlled operations within the poultry farmhouse. By providing real-time data, the system enables better decision-making, automates processes, and offers a more efficient and cost-effective approach to managing and monitoring operations. This integration of sensors and actuators in an IoT framework empowers the poultry farm to optimize temperature control and other environmental factors, ensuring optimal conditions for the growth and health of the chicks while minimizing manual intervention.

### Relative humidity (RH)

Relative humidity is a pivotal parameter that determines the moisture content present in the air and holds significant importance in diverse applications, including poultry farming. Equation (1) provides a precise method for calculating relative humidity ($\phi$) by comparing the partial pressure of water vapor ($p_{H_2O}$) with the equilibrium vapor pressure of water ($p^*_{H_2O}$).

$$\phi = \frac{p_{H_2O}}{p^*_{H_2O}} \tag{1}$$

The IoT system introduced in this study incorporates the SHT20 humidity sensor to effectively monitor and evaluate relative humidity levels within the poultry farm. By utilizing Eq. (1), the humidity sensor provides accurate readings that facilitate the calculation of relative humidity. To align with this study's focus, the collected relative humidity values are transformed into a percentage scale. The resulting dataset is then visualized in Fig. 8, offering insights into the daily average relative humidity levels over a specified timeframe. In this graphical representation, the X-axis corresponds to the dates, while the Y-axis denotes the percentage values of relative humidity.

Sustaining an optimal relative humidity level significantly influences the well-being and growth of poultry. Conventionally, a relative humidity level of 80% is deemed ideal for

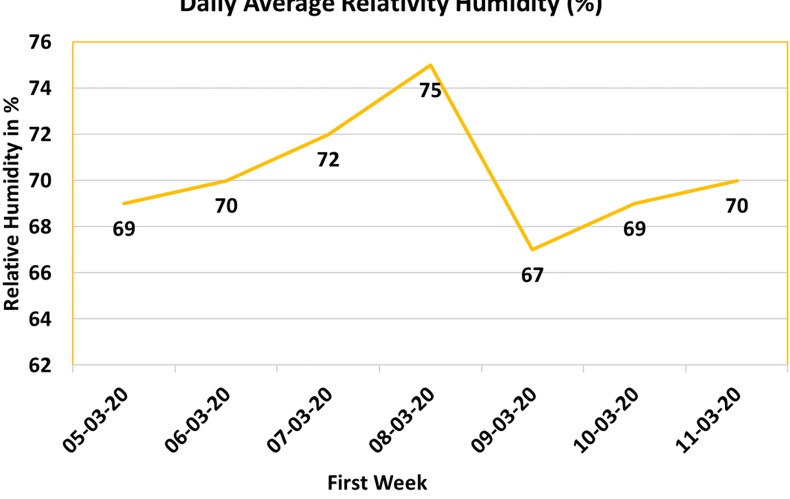

**Figure 8** Relative humidity (RH%) value.

poultry farming. However, the observations during the first week unveiled that the average relative humidity values consistently hovered around 70%, as depicted in Fig. 8.

To ensure that relative humidity conditions remain conducive for the chickens, the IoT system is intelligently designed to generate notifications for the farm supervisor if the relative humidity percentage falls below a predefined threshold. Upon receiving such notifications, the supervisor can promptly initiate corrective measures, such as activating a humidifier, to uphold the required humidity percentage. This proactive approach guarantees that the poultry farm promptly addresses any deviations from the optimal humidity levels, thereby promoting the overall health and growth of the avian inhabitants.

### Oxygen sensor ($O_2$)

Maintaining an optimal level of oxygen concentration within the confines of poultry farm control sheds is of paramount importance to avert congestion-related concerns and foster the healthy development of young chicks. As emphasized by *Saleeva et al. (2020)*, sustaining an oxygen concentration of 21% volume (Vol) is deemed ideal for ensuring optimal growth within poultry farms. To gauge the oxygen concentration, the deployed system integrates the Grove $O^2$ sensor. The resultant dataset is then utilized to compute the daily average, subsequently visualized in Fig. 9. In this graphical representation, the X-axis corresponds to dates, while the Y-axis depicts the oxygen concentration in terms of percentage volume.

Upon scrutinizing the dataset showcased in Fig. 9, it is evident that the early weeks witnessed relatively lower oxygen concentration levels. However, with the passage of time, a noteworthy enhancement is discernible, with oxygen concentration levels approaching the vicinity of 18%. This progression aligns more closely with the recommended oxygen concentration level posited by *Saleeva et al. (2020)*. Importantly, maintaining a higher oxygen concentration within the control sheds contributes to a more favorable and comfortable environment for the chicks, thereby fostering superior growth rates. This

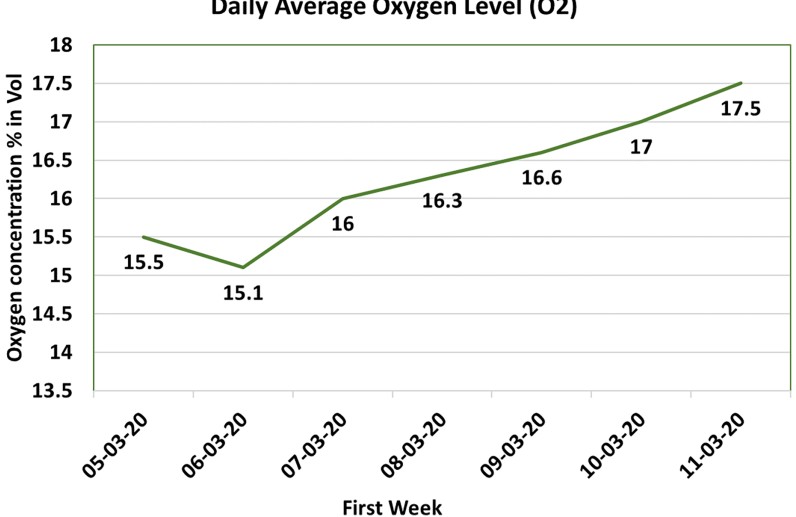

**Figure 9 Oxygen ($O_2$) value.**

underscored the significance of adequate oxygen levels underscores their pivotal role in promoting the overall health and well-being of the avian occupants.

Drawing insights from the system-generated data, a critical threshold of 16% oxygen concentration was identified. To ensure that oxygen levels consistently exceed this critical threshold, the system is adeptly configured to initiate an SMS notification to the farm supervisor whenever the oxygen concentration registers a dip below this prescribed minimum. This notification serves as an alert, compelling the supervisor to implement requisite measures, such as optimizing ventilation or augmenting airflow, to promptly reestablish and sustain the requisite oxygen concentration levels. Through the continuous vigilance and regulation of oxygen concentration levels within the control sheds. The poultry farm is empowered to provide an environment conducive to healthy and optimal chick development. By averting congestion-related challenges and ensuring oxygen-rich surroundings, the poultry farm can ensure unfettered growth and well-being for its avian inhabitants.

### Carbon dioxide sensor ($CO_2$)

Carbon dioxide ($CO_2$) gas is an inherent byproduct of chick respiration. Monitoring and regulating $CO_2$ levels within poultry farm environments is imperative, as elevated concentrations can adversely affect the respiratory health of the avian inhabitants. While typical ambient conditions yield $CO_2$ levels of approximately 450 parts per million (ppm) (*Chung, 2021*), it remains pivotal to ensure that the maximum $CO_2$ level within the poultry farm remains below 2,500 ppm to preempt respiratory issues among the chicks.

The measurement of $CO_2$ gas concentration within this system leverages the MQ-135 sensor. Subsequent to data collection, a daily average is computed, subsequently rendered in Fig. 10. In this graphical depiction, the X-axis delineates the dates, while the Y-axis denotes the $CO_2$ gas concentration expressed in ppm. Notably, Fig. 10 chronicles the peak $CO_2$ gas concentration within the poultry farm, which approached 1,100 ppm during the

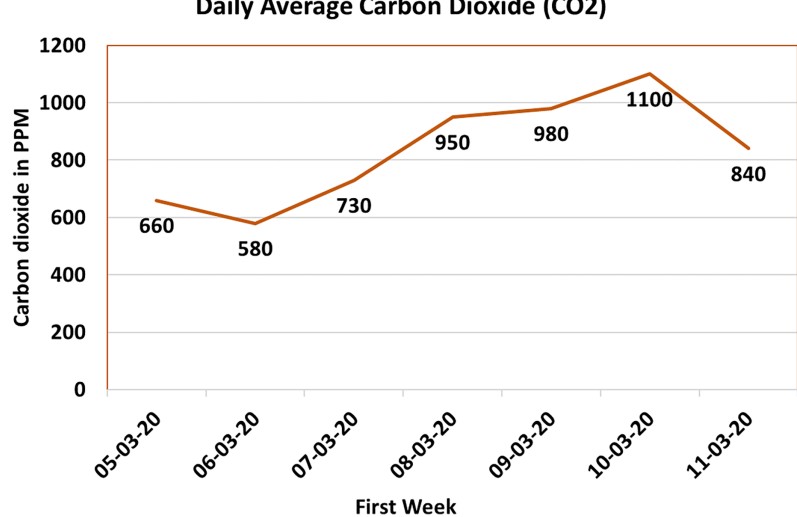

**Figure 10 Carbon dioxide ($CO_2$) value.**

documented period. It is imperative to underscore that this recorded timeframe coincides with a period of shed closure aimed at maintaining the requisite temperature. The escalated $CO_2$ levels observed in confined settings underscore the significance of efficacious ventilation and robust monitoring systems to obviate the accumulation of noxious gases.

In the pursuit of timely intervention and the perpetuation of a secure environment for the chicks, the system is astutely configured to dispatch an SMS notification to the farm supervisor upon the $CO_2$ gas level surpassing the 500 ppm threshold. This demarcation operates as an early alert, propelling the supervisor to undertake judicious measures, such as calibrating ventilation systems or instituting supplemental protocols to attenuate the $CO_2$ concentration. Through the continuous oversight and management of $CO_2$ gas levels, the poultry farm is primed to provide a salubrious and congenial milieu for the chicks. This concerted approach mitigates the specter of respiratory ailments and concomitantly nurtures the holistic well-being of the avian occupants.

### Carbon monoxide sensor (CO)

Carbon monoxide (CO) gas, characterized by its colorless and odorless nature (*Ibrahim, 2018*), poses a profound threat due to its highly toxic attributes. Predominantly encountered as an industrial hazard, CO gas carries substantial implications for both human and animal well-being. Within the precincts of a poultry farm, CO gas can emanate from diverse sources, including the operation of diesel generators within the shed or the decomposition of manure generated by the chicks.

Figure 11 captures the measurements of CO gas levels amassed through the orchestrated system, facilitated by the Winson ZEO7-CO sensor. The data is succinctly depicted as the average daily concentration of CO gas, conveyed in parts per million (ppm). In this graphical portrayal, the X-axis delineates the temporal evolution, while the Y-axis signifies the CO gas concentration in ppm.

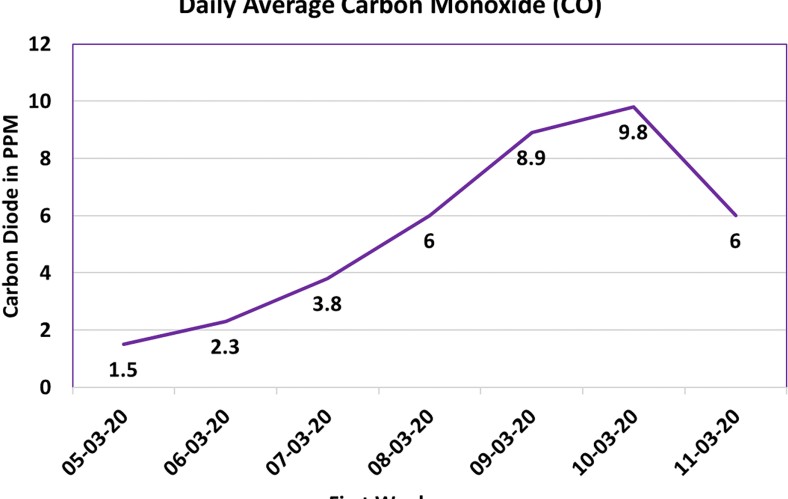

**Figure 11 Carbon monoxide (CO) value.**

Analyzing Fig. 12, it is discernible that the CO gas concentration exhibited an initial period of dormancy, subsequently ascending to a pinnacle of 10 ppm, only to descend towards the culmination of the observed week. This oscillatory pattern underscores the potential interplay of multiple sources and conditions precipitating CO gas generation within the poultry farm milieu.

The criticality of maintaining CO gas concentrations at safe and acceptable levels cannot be overstated, particularly given its perilous ramifications, especially for susceptible chicks. Prolonged exposure to elevated CO gas levels engenders severe health consequences (*Ibrahim, 2018*). Consequently, the surveillance and regulation of CO gas levels within the poultry farm environment are indispensable.

In the overarching endeavor to safeguard the well-being of the chicks, the IoT system assumes a pivotal role by effectuating continuous CO gas level monitoring. This perpetual vigilance facilitates the timely discernment of substantial deviations, thereby enabling prompt interventions and ameliorative measures to avert potential harm to the avian occupants.

### Ammonia (NH₃) gas sensor

Ammonia ($NH_3$) gas, characterized by its pungent odor, arises as a byproduct of manure decomposition within poultry farmhouses. The presence of ammonia in such environments engenders significant risks to the health and welfare of chickens. Exposure to elevated ammonia concentrations, such as 80 parts per million (ppm), can precipitate a spectrum of health issues, ranging from ocular ulcers to severe respiratory tract impairment (*Naseem & King, 2018*). These afflictions culminate in discomfort, irritation, and potentially respiratory distress for the avian inhabitants.

The meticulous surveillance of ammonia gas concentration within poultry farmhouses is undertaken through an advanced sensing paradigm. The deployed system incorporates a Winson-ME3NH3 sensor, meticulously designed to gauge and monitor ammonia gas

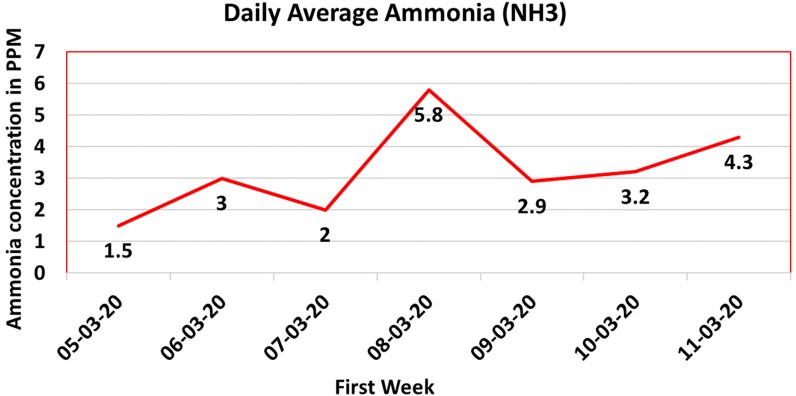

**Figure 12  Ammonia gas (NH₃) value.**

levels accurately. The ensuing dataset is subjected to comprehensive analysis and subsequently translated into a graphical representation, capturing the daily average ammonia gas concentration over time. Figure 12 furnishes an illustrative rendition of this measured data, wherein the temporal progression is depicted along the X-axis, while the ammonia gas concentration in parts per million (ppm) is charted along the Y-axis.

The pivotal implication conveyed by Fig. 12 is the sustenance of ammonia levels well below the prescribed upper limits, a fundamental prerequisite for upholding a salubrious environment for poultry. The continuous monitoring orchestrated by the IoT system assures expeditious identification and mitigation of potential issues.

To further fortify safety protocols within poultry farmhouses, the IoT framework incorporates an automatic mechanism. Upon surpassing the 80 ppm threshold, a high-priority alert is instantaneously disseminated to both the supervisor and the farm's medical personnel. This expeditious notification mechanism facilitates timely intervention and prudent remedial measures to counteract ammonia accumulation, thereby safeguarding the well-being and vitality of the avian occupants.

## Discussion and lesson learned

This section details the valuable experience we have gained during the implementation of the PEMS system in the field. As we can see from the results presented in "Result", environmental parameters contribute heavily to the health and the growth of the birds and thus to the profitability of the farm. Monitoring live environment parameters and implementing associated alert systems can ensure healthy growth to optimize the productivity of a chick farm. More importantly, by live monitoring of gases, such as $NH_3$, $CO_2$, and CO, a farmer can reduce the level of gases that are very harmful to the health of birds. This system has revealed some lessons that can be useful for future smart farming practices in chicken farming. These lessons are summarized below.

- The environment within the control sheds of the poultry farms is dusty all the time. However, the dust particles will impact the ageing and performance of sensors, controllers, and electronic components. Thus, the proper protection and timely cleaning of the devices themselves are very important.

- The placements of sensors are very important because in control sheds every corner has different temperature and humidity levels. Temperature and humidity extremes occur at the pad side (low-temperature side) and the fan side (dry area).
- For accurate measurement of the data, strategic sensor placement with reliable data interpretation and analysis is needed.
- The reading range of LoRa modules is affected by the civil structure of the shed because it is shaped like a tunnel. Thus, modules with antennae are better options.

## LIMITATIONS AND FUTURE ROADMAP

In this section, we discuss the limitations of this study and also provide a future roadmap to enhance this work.

### Limitations

Despite the promising findings and contributions of this study, some limitations should be acknowledged.

Firstly, the data collection was limited to a span of 1 week, which may not capture long-term variations or seasonal effects. Secondly, the study focused on a specific poultry farm, and the results may not be directly generalizable to other farms with different setups or geographic locations. Variations in farm size, ventilation systems, and management practices could influence the observed sensor readings. Thirdly, while the IoT-based sensor systems provided valuable real-time data, they were limited to monitoring and alerting functionalities.

### Future roadmap

Building upon the limitations and opportunities identified, several avenues for future research and development can be explored.

Firstly, expanding the study to encompass multiple poultry farms would provide a broader understanding of the variations and challenges across different farm settings. Comparing and analyzing data from various farms could lead to the identification of best practices and guidelines for optimal environmental management (*Murugeswari et al., 2023*).

Secondly, incorporating machine learning and data analytics techniques (*Castro et al., 2023*; *Yang et al., 2023*) could enhance the capabilities of IoT-based sensor systems. By leveraging advanced algorithms, it would be possible to detect patterns, predict potential deviations, and recommend proactive interventions to maintain optimal environmental conditions.

Furthermore, integrating the sensor systems with automated actuators and control mechanisms would enable real-time adjustments based on the sensor readings (*Mazunga et al., 2023*). This would create a more intelligent and self-regulating farm environment, reducing the reliance on manual interventions and optimizing resource utilization. Besides, utilizing energy harvesting techniques (*Karim et al., 2018*) for the longevity of IoT nodes may also be recommended.

Additionally, exploring the integration of other relevant environmental parameters, such as light intensity and air quality, would provide a more comprehensive understanding of the poultry farm environment. These additional parameters can contribute to a holistic approach to environmental management and further enhance animal welfare and productivity (*Ramteke & Dongre, 2023*).

Finally, long-term studies encompassing multiple seasons and variations in weather conditions would provide a more complete understanding of the environmental dynamics in poultry farms (*Jacobs et al., 2023*). This would enable farmers to develop targeted strategies and interventions to mitigate potential risks and optimize conditions for the birds' health and growth. Autonomous vehicles can also be utilized to augment the production of poultry farms (*Shi et al., 2023*).

## CONCLUSIONS

This study focused on implementing an advanced technology-based system for the efficient management of chickens in poultry farms by continuously monitoring environmental parameters. The continuous monitoring of temperature, RH, O, $CO_2$ levels, CO levels, and $NH_3$ gas concentration, allowed for a comprehensive understanding of the farm's environmental conditions.

The findings highlighted the importance of maintaining appropriate levels of these parameters to ensure the health and well-being of the chicks. The temperature remained within the desired range, ensuring a comfortable environment for the chicks without exceeding the recommended upper limit. The $O_2$ concentration levels showed a gradual improvement over the entire period of monitoring, but occasional drops below the minimum threshold necessitated immediate action to restore optimal conditions for the birds' growth and development. The consistently low levels of $NH_3$ gas and CO indicated a safe respiratory environment, while occasional fluctuations in RH and $CO_2$ levels emphasized the need for consistent monitoring and management.

The implementation of IoT-based sensor systems offers significant advantages in poultry farm management, enabling real-time data collection, improved decision-making, and automated processes. Further enhancements could involve integrating these systems with actuators for precise control of environmental conditions, further optimizing the growth and health of the poultry.

Overall, this study demonstrates the potential of IoT technologies in enhancing poultry farm management, promoting animal welfare, and improving operational efficiency. By harnessing the power of sensors and IoT connectivity, poultry farmers can create a safer, healthier, and more sustainable environment for their birds.

## NOMENCLATURE

| | |
|---|---|
| **AoI** | Age of Information |
| **AWS** | Amazon Web Server |
| **CO** | Carbon monoxide |

| | |
|---|---|
| $CO_2$ | Carbon dioxide |
| $NH_3$ | Ammonia |
| DRL | Deep Reinforcement Learning |
| EECO | Energy Control and Computation Offloading |
| GHz | Giga Hertz |
| GPRS | General Packet Radio Services |
| GSM | Global System for Mobile Communication |
| GUI | Graphical User Interface |
| I2C | Inter-Integrated Circuit |
| IBM | International Business Machines |
| IIoT | Industrial Internet of Things |
| IoT | Internet of Things |
| JDK | Java Development Kit |
| LCD | Liquid Crystal Display |
| LED | Light Emitting Diode |
| LoRa | Long Range Radio |
| LoRaWAN | Long Range Wide Area Network |
| LPWAN | Low Power Wide Area Networks |
| LSI | Large-Scale Integration |
| MDP | Markov Decision Process |
| MEMS | Micro-Electromechanical Systems |
| MQTT | Message Queuing Telemetry Transport |
| MySQL | My Structured Query Language |
| NB-IoT | NarrowBand-Internet of Things |
| PC | Personal Computer |
| PEMS | Portable Emissions Measurement System |
| PLC | Programmable Logic Controller |
| ppm | parts per million |
| RH | Relative Humidity |
| SaaS | Software as a System |
| SIM | Subscriber Identity Module |
| SMS | Short Messaging Services |
| URL | Uniform Resource Locator |
| Vol | Volume |
| VR | Ventilation Rate |
| Wi-Fi | Wireless Fidelity |
| WLAN | Wireless Local Area Network |
| WSN | Wireless Sensor Network |

### Funding

This research is funded by the Researchers Supporting Project Number (RSPD2023R947), King Saud University, Riyadh, Saudi Arabia. The funders had no role in study design, data collection and analysis, decision to publish, or preparation of the manuscript.

### Grant Disclosures

The following grant information was disclosed by the authors:
Researchers Supporting Project Number: RSPD2023R947.

### Competing Interests

Khursheed Aurangzeb is an Academic Editor for PeerJ.

### Author Contributions

- Muhammad Hanif Lashari conceived and designed the experiments, performed the experiments, analyzed the data, performed the computation work, prepared figures and/or tables, and approved the final draft.
- Sarang Karim conceived and designed the experiments, performed the experiments, performed the computation work, prepared figures and/or tables, and approved the final draft.
- Musaed Alhussein analyzed the data, authored or reviewed drafts of the article, and approved the final draft.
- Ayaz Ahmed Hoshu performed the experiments, analyzed the data, authored or reviewed drafts of the article, and approved the final draft.
- Khursheed Aurangzeb conceived and designed the experiments, performed the experiments, performed the computation work, prepared figures and/or tables, and approved the final draft.
- Muhammad Shahid Anwar analyzed the data, authored or reviewed drafts of the article, and approved the final draft.

### Data Availability

 The source code and dataset are available in the Supplemental Files.

### Supplemental Information

Supplemental information for this article can be found online at http://dx.doi.org/10.7717/peerj-cs.1623#supplemental-information.

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
