# Peer review of "Internet of Things-based sustainable environment management for large indoor facilities"

_PeerJ Computer Science, doi:10.7717/peerj-cs.1623_

## Round 0.1 · original submission · Major Revisions

The paper needs more work in terms of explaining by adding more details and make things clear, expose the novelty to the reader is very important.

Reviewer 1 ·

Basic reporting

1. The research background still needs in-depth analysis.
2. need more to discuss your results.
3. Suggest to introduce more references in the discussion to compare and analyze your results.
DOI: 10.1109/TMC.2023.3240763;https://doi.org/10.1016/j.future.2022.09.007; https://doi.org/10.3390/s23031695
4. The conclusion section needs to focus on the practical effectiveness of the innovation point of the article.
5. What are the shortcomings of the study and future prospects.

Experimental design

The authors should provide more details on how they validated the effectiveness and robustness of the algorithm.

Validity of the findings

Please provide more details.

Additional comments

none

·

Basic reporting

Even article is written in clear English and provide certain literature analysis, it main drawback is missing novelty. There are multiple articles published, not referenced here about environment factor measurements with sensors in the buildings where animals are kept. Even particularly where chickens are kept. For example, Arhipova I. et. al. Smart Platform Designed to Improve Poultry Productivity and Reduce Greenhouse Gas Emissions (2021). etc.

Software solution described in this article is not clear. Page 9 states "various tools are used". Article would benefit from clear description for software development technology stack used and implementation of the software. It would be interesting to see data base model for storing all the sensor data in a scalable manner.

Article figures and tables may need to be better quality. For example Figure 2 is large, but does not include much information for the size. Names of multiple figures do not describe what the reader can actually see in the figures. Is Figure 1 really a model? In Table 2 reader can find only sensor model names not much details about sensors. And Is this table necessary? Names and descriptions are already included in the text before.Figure 2 really shows placement of the sensors. Wouldn't it be more beneficial so see actual photo from the placements or more detail of sensor locations in the building?

Experimental design

Even article fits the scope of the journal, there is not enough description on experimental setup. Reader can find sensors and protocols used, but software description is weak as well as placement of the sensors.

Validity of the findings

Findings are very broad. For example "Strategic sensor placement with reliable data interpretation and analysis is needed" is conclusion found in many similar articles as well as finding that sensors placed in different places in the building will provide different results. It is well known fact from climate control inside buildings. Same with the finding that dust will impact sensor longevity.
Readers would benefit from findings about sensor interactions with animals, need for particular sensors, count of sensors for precision, guidelines for multiple sensor data aggregation, etc.

Additional comments

There are still some grammar mistakes and reference mistakes. "Sousa" is named 2 times (page 3). "Jacome" is misspelled (page 3). "Temprature" spelling (Figure 2). Full stop two times Page 9 line 317. Figures with graphs need explanation what reader can see on X axis, etc.

---

## Round 0.2 · Minor Revisions

Please address the remaining minor revisions from Reviewer 2

·

Basic reporting

Improvements can be seen in literature references and some background information. I do think that charts included in the article are rather small in size and can be improved for visibility.

Experimental design

There are still statements by authors to include certain things in following articles, such as description of sensors and description of particular constraints such as animal interaction with sensors and impact on precision, etc. With that taking in consideration present version of the article can be satisfactory.

Validity of the findings

Novelty assessment is still partially questioned, however there is improvement with second version of the article.

Additional comments

I think with slight improvements in charts article could be considered for publication. Figures with charts also ask clarification about x axis - assumed those are dates?
There are still spelling issues - Celsius spelling in Figures are incorrect. References to Figure X is in the text. Description of what can be seen in Figure 6 would add value.

Section 3.1.2.is still include - "various tools are used". It would be appropriate to state for which tasks which tools were used. For example Wild Fly is a product name, but authors state as " wildfly application server". Clarity in that section is still needed so readers can understand tools and purpose for the tools used. For example authors use "Excel format". Microsoft Excel is a product which supports opening and storing of various formats - xls, xlsm to start with.

I think article should be checked again for grammar and use of professional language to describe software and hardware solutions used. Figure 3 could be more informative and better structured.

---

## Round 0.3 · accepted · Accept

The paper has been edited very well, and all the commnets provided by the reviewresr are addreesed by the authors. So I am happy to accept this paper.

Reviewer 1 ·

Basic reporting

All my previous concerns have been addressed.

Experimental design

good

Validity of the findings

Good

Additional comments

The authors have revised the problems. I recommend that this paper should be accepted.